# Factors supporting and constraining the implementation of robot-assisted surgery: a realist interview study

Rebecca Randell,[1] Stephanie Honey,[2] Natasha Alvarado,[1] Joanne Greenhalgh,[3] Jon Hindmarsh,[4] Alan Pearman,[5] David Jayne,[6] Peter Gardner,[7] Arron Gill,[8] Alwyn Kotze,[9] Dawn Dowding[10]

For numbered affiliations see end of article.

**Correspondence to**
Dr Rebecca Randell;
R.Randell@leeds.ac.uk

## ABSTRACT

**Objective** To capture stakeholders' theories concerning how and in what contexts robot-assisted surgery becomes integrated into routine practice.

**Design** A literature review provided tentative theories that were revised through a realist interview study. Literature-based theories were presented to the interviewees, who were asked to describe to what extent and in what ways those theories reflected their experience. Analysis focused on identifying mechanisms through which robot-assisted surgery becomes integrated into practice and contexts in which those mechanisms are triggered.

**Setting** Nine hospitals in England where robot-assisted surgery is used for colorectal operations.

**Participants** Forty-four theatre staff with experience of robot-assisted colorectal surgery, including surgeons, surgical trainees, theatre nurses, operating department practitioners and anaesthetists.

**Results** Interviewees emphasised the importance of support from hospital management, team leaders and surgical colleagues. Training together as a team was seen as beneficial, increasing trust in each other's knowledge and supporting team bonding, in turn leading to improved teamwork. When first introducing robot-assisted surgery, it is beneficial to have a handpicked dedicated robotic team who are able to quickly gain experience and confidence. A suitably sized operating theatre can reduce operation duration and the risk of de-sterilisation. Motivation among team members to persist with robot-assisted surgery can be achieved without involvement in the initial decision to purchase a robot, but training that enables team members to feel confident as they take on the new tasks is essential.

**Conclusions** We captured accounts of how robot-assisted surgery has been introduced into a range of hospitals. Using a realist approach, we were also able to capture perceptions of the factors that support and constrain the integration of robot-assisted surgery into routine practice. We have translated these into recommendations that can inform future implementations of robot-assisted surgery.

## INTRODUCTION

Laparoscopic surgery provides benefits for patients, including less postoperative pain, shorter hospitalisation, quicker return to normal function and improved cosmetic effect.[1–3] However, it can be technically

### Strengths and limitations of this study

► This is the first study to provide detailed insight into stakeholders' views of robot-assisted surgery implementation.
► Interview questions were based on analysis of existing literature, which enabled us to explore the extent to which findings from single site studies were more widely applicable.
► We interviewed the full range of operating theatre personnel, enabling us to add to and refine the literature-based theories to reflect the experience of a broad range of stakeholders.
► A limitation is that interviews were conducted with staff from only one surgical specialty, limiting generalisability, although the theories that were explored in the interviews were derived from literature concerning a range of surgical areas.
► While we report staff perceptions of the factors that support and constrain the integration of robot-assisted surgery, the resulting theories remain to be empirically tested.

challenging to perform, due to the two-dimensional image of the surgical site and instruments that have limited freedom of movement and require awkward and non-intuitive handling, resulting in slow uptake.[4] The da Vinci robot (Intuitive Surgical, California, USA), a master-slave (or console-manipulator) system,[5] aims to reduce these challenges. The robot provides a stable camera image with a three-dimensional image of the surgical site, intuitive instrument handling, tremor elimination, motion scaling and instruments with increased freedom of movement. Clinical evidence of patient benefits have led National Health Service (NHS) England to recommend use of robot-assisted surgery for radical prostatectomies[6] and treatment of early stage kidney cancer,[7] although uncertainty regarding the benefits for other operations remains.

The latest model of the da Vinci robot costs about £1.7m, with annual maintenance fees of about £140 000 per robot.[8] Given these high costs, with the cost effectiveness of robot-assisted surgery depending on the number of operations for which the robot is used,[9] it could be anticipated that hospitals that have purchased a da Vinci robot would be seeking to maximise use. However, implementation of robot-assisted surgery can be challenging, with reports of da Vinci robots being introduced but then underused.[10] While accounts of the introduction of robot-assisted surgery suggest a number of factors important for successful integration, they come from small case series undertaken in single institutions, typically by dedicated robot-assisted surgery enthusiasts,[3] so little is known about the contextual factors necessary for successful integration of robot-assisted surgery more broadly.

In this paper, we report the results of an interview study that was undertaken with the purpose of providing guidance to healthcare organisations that are considering the introduction of robot-assisted surgery or are seeking to increase use of robot-assisted surgery. We sought to answer the following question: how and in what contexts does robot-assisted surgery become integrated into practice?

## METHODS

The interview study was conducted as part of a process evaluation that ran alongside RObotic versus LAparoscopic Resection for Rectal cancer (ROLARR), a multicentre, randomised controlled trial comparing laparoscopic and robot-assisted surgery for the curative treatment of rectal cancer.[11 12] Realist evaluation, which involves eliciting, testing, and refining stakeholders' theories of how an intervention works, provided an overall framework for the process evaluation.[13] Realist evaluation was considered appropriate for studying the integration of robot-assisted surgery because it has been used for studying the implementation of a number of complex interventions in healthcare[14–17] and because it explicitly acknowledges the sociotechnical nature of technologies

such as robot-assisted surgery. For realists, technologies offer resources to recipients and the outcomes depend on recipients' responses to those resources, which are likely to vary according to the context into which the technology is introduced. This combination of resources and recipients' responses are understood as the mechanisms through which a technology achieves its outcomes.[18] The question asked is not 'does the technology work?' but 'what works, for whom, under what conditions and how?' Consequently, realist theories are expressed in the form of Context Mechanism Outcome configurations, where Context+Mechanism = Outcome.

The first stage in realist evaluation is eliciting stakeholders' theories about how the intervention works,[19] using strategies such as identifying relevant theories from the literature, reviewing the existing literature on the topic or interviewing stakeholders. We used a combination of these approaches. A review of literature evaluating how and in what contexts robot-assisted surgery becomes integrated into practice was used to develop a series of tentative theories,[20] which are summarised in table 1. These theories were then refined through interviews with operating theatre (OT) teams.

### Setting and participants

Ten English NHS hospitals were using robot-assisted surgery for colorectal surgery at the time of the interviews. We invited OT teams in all 10 hospitals to participate in the interview study, ensuring the OT teams involved in the study varied in their level of experience with robot-assisted surgery. To capture the perspectives of all professional groups that make up the OT team, a snowball sampling strategy was used.[21] At each hospital, one of the surgeons was interviewed first and he or she then helped to identify other OT team members to interview.

### Data collection

Data collection and analysis was undertaken by three experienced qualitative researchers (RR, SH and NA), one of whom (SH) is a registered nurse. Semi-structured interviews were conducted by telephone, employing a

| Context | + | Mechanism | | = | Outcome |
|---------|---|-----------|---|---|---------|
| | | Resource | Response | | |
| Support of hospital administration and nursing management | + | Additional staff | Assist with setting up and clearing away robot | = | Reduced set-up time |
| Availability of additional staff with experience of robotic set-up | | | | | Quicker turnover to next case |
| Motivated and stable team | + | Dedicated robotic team | Team sees operations as opportunity to learn and more quickly become familiar and confident with equipment and tasks | = | Reduced set-up time |
| High number of frequent robotic operations | | | | | |
| Support of hospital administration and nursing management | | | | | |
| Support of hospital administration and nursing management | + | Dedicated robotic OT | Team does not need to move robot from/to another location before/after operation | = | Reduced set-up time |
| Availability of suitably sized operating theatre (OT) | | | | | Quicker turnover to next case |

Table 1  Tentative theories from the literature review

realist technique called the teacher/learner cycle.[22] Participants were presented with the literature-based theories and asked to reflect on whether, and in what ways, those theories fitted with their own experiences and to refine or modify these ideas accordingly. While such an approach is very different to a typical qualitative interview where the interviewer is expected to put aside any preconceptions or assumptions, realists argue that the interviewer always has their own theories when going into an interview, which influences the questions they ask and how they ask them, and similarly the interviewee always has their own ideas about what the interviewer is interested in, which influences the answers they provide. Therefore, in theory-driven research, a more productive approach is to use the interview as a vehicle for enabling key participants to revise and expand these theories.

Interviews were audio recorded and transcribed verbatim. An iterative approach to data collection and analysis was taken and the interview topic guide was revised as new theories and revisions to the theories were identified.

All participants gave informed consent; because the interviews were undertaken by telephone, consent was verbal rather than written.

## Data analysis

Following anonymisation, interview transcripts were analysed using framework analysis.[23] Codes used for indexing the data focused on capturing how our initial theories were expanded, supported and refined and how different contexts shaped the mechanisms through which robot-assisted surgery was perceived to become integrated into practice. The indexed data were summarised in a matrix display to build up a picture of the data as a whole,[24] supporting both comparisons within a single hospital and comparisons between hospitals. Finally, refined theories were developed through a process of discussing narrative summaries of the indexed data, comparing findings with the tentative literature-based theories.

## Patient involvement

A lay member who was part of the research team contributed to study design and management and provided a patient perspective on analysis and interpretation of the data. A Patient Panel chaired by the lay member provided advice on selection of key theories for testing in later phases of the research and on appropriate strategies for disseminating research findings to relevant interest groups.

## Participants' characteristics

Nine of the 10 hospitals approached agreed to participate. We conducted semi-structured interviews with 44 staff, covering a range of professional groups (see table 2). Interviews ranged from 29 min to 1 hour 40 min, with an average (mean) length of interview of 53 min.

**Table 2** Participants by professional group and hospital type

| n=44 | n (% of sample) |
| --- | --- |
| Professional group | |
| Surgeon | 12 (27) |
| Surgical trainee | 5 (11) |
| Manager | 1 (2) |
| Anaesthetist | 6 (14) |
| Nurse | 13 (30) |
| Operating department practitioner | 7 (16) |
| Hospital type | |
| Teaching | 21 (48) |
| District general | 17 (39) |
| Cancer centre | 6 (13) |

## Organisational support

The literature review revealed that robot-assisted surgery introduces challenges that can constrain its use. A key issue is that it can extend operation duration, although this effect reduces with experience.[25–27] Consequently, support of the hospital administration and nursing management is necessary for the integration of robot-assisted surgery, to ensure provision of adequate resources, such as additional OT time.[28 29] How to obtain support was not explicated in the literature, although the need to create a 'shared vision' of what the introduction of robot-assisted surgery would enable was described.[30] The tentative theory explored in the interviews was that, where hospital administration and nursing management are involved in the decision to introduce robot-assisted surgery, they will perceive the potential benefits of robot-assisted surgery as assisting in achieving the organisation's goals and will be willing to invest resources, such as additional staff, to support its integration into practice. Our participants agreed with this theory. They identified the support of hospital administration as important because of the possible negative consequences of the longer operation duration and the impact this could have on waiting lists. Consequently, surgeons would not accept responsibility for implementation of robot-assisted surgery without support from the hospital administration.

Participants also provided insight into some of the ways in which support was achieved. Creating a shared vision in some cases literally meant giving the hospital administration the opportunity to see the robot in action:

> They came and watched a full case and I talked to them afterwards and they said it was very, very informative to actually see what goes on compared to what they hear. And to actually see it they realised how impressive it was and also the benefits to the patient […] I got a lovely email off both of them saying it was very informative and […] when they can go to the board of management […] they can then have a

better idea of what they're talking about to promote robotic surgery. (Site 4, ODP)

While this quote emphasises the perception of patient benefits, other participants emphasised the hospital administration's awareness of competition, which could outweigh concerns about cost. Robot-assisted surgery was perceived as a mark of prestige and enabled hospitals to be viewed as providing cutting edge services, which in turn enhanced the likelihood such services would be retained:

I think the fact that we were the first in this part of the country to have it. […] It was considered a very prestigious move, so yes it was considered, you know, to be such a futuristic addition to our theatres that it was very exciting. (Site 5, Nurse)

When asked about the role of nursing management, participants talked instead about the importance of team leaders, a role taken on by experienced theatre nurses and operating department practitioners (ODPs). A supportive team leader could facilitate integration by:

► Gaining access to training for team members, which contributed to safety and to confidence in using the equipment.
► Co-ordinating staff rotas to ensure the right skill mix was available to carry out robot-assisted operations.
► Co-ordinating robot use across specialties to maximise use.
► Managing OT schedules to allow, at least initially, for longer set-up times and for availability of an OT suitable to accommodate equipment and personnel safely, without risk of de-sterilisation or compromising patient access.

Finally, support from surgical colleagues was perceived as important. As one participant explained:

You need the absolute support of your [surgical] colleagues…First of all if you're going to start spending […] all day lists on your first ten cancers then your waiting list increases or the pressure on others increases. If there's any murmuring from the background…you will start to avoid doing this [robot-assisted surgery]. Secondly if colleagues hate the idea of others learning a skill or getting a reputation which they don't have yet, they could scupper this happening. I've been lucky that those things don't count here and that's one of the reasons why I can progress. When I speak to colleagues they cite one or all of those, say they're not actually allowed to progress. (Site 7, Surgeon)

### Dedicated team
The literature review also identified strategies used by OT teams to reduce operation duration and thereby support integration of robot-assisted surgery. One strategy was to have a dedicated robotic team[28][30–37] who can 'work through the learning curve and, if possible, all robotic

cases'.[38] Factors that impact effectiveness of this strategy are number and frequency of robot-assisted operations and team motivation[30] and stability.[39] The tentative theory discussed in the interviews was, where there is a motivated and stable team and a high number of frequent robot-assisted operations, a dedicated robotic team will see operations as an opportunity to learn and will more quickly become familiar and confident with equipment and tasks, leading to a reduced set-up time. Participants agreed with this theory and reported that, in many cases, people who trained together became a dedicated robotic team, at least initially:

When we had a dedicated team of people who could manoeuvre the robot and position patients…to start with you do need a core knowledge…it definitely did reduce the time having the same skill set. (Site 4, Surgeon)

However, it was not always possible to maintain a dedicated team due to staff changes, holidays and sickness. Theatre nurses and ODPs often only work within one or two specialties, making it hard to achieve a dedicated team, especially where there was a low volume of robot-assisted cases. Where a dedicated team was not feasible, a larger pool of people, trained by experienced staff, was established. At some sites 50 per cent of the staff had been trained and at one site, which carried out a large volume of robot-assisted cases, all staff could manage the cases.

### Dedicated operating theatre
Another strategy to reduce operation duration and thereby support the integration of robot-assisted surgery into practice was having a dedicated robotic OT.[28 40] This means the robot does not need to be moved between OTs, reducing time spent setting up and putting away the robot. Participants agreed with this and, while only three sites had a dedicated OT, participants felt a dedicated OT would be the ideal situation. Where there was not a dedicated OT, team leaders were perceived to play a vital role in ensuring a suitably sized OT was available. Participants felt a suitably sized OT would make robot-assisted surgery more efficient because a cramped working environment meant staff struggled to move around quickly and safely. It could also lead to accidental de-sterilisation of equipment, with implications for patient safety and, because it is then necessary to replace or re-drape the equipment, operation duration and costs.

### Implementation processes
In addition to refining the literature-based theories, we also captured participants' accounts of how robot-assisted surgery was introduced into their hospital, to identify differences in implementation strategies between sites. This identified additional theories about what is required for integration of robot-assisted surgery.

## Whole team training

Approaches to OT team training varied significantly between sites. There was also variation within sites, depending on role and at what point in time OT personnel joined the hospital. Participants who had undertaken training as a team suggested the important aspect of training was that it enabled them to develop trusting relationships with each other, which in turn allowed them to work together to solve problems arising from the implementation of robot-assisted surgery. Teams that had undertaken training together in an Intuitive Surgical training centre said it was 'inspiring' and had a 'bonding' effect:

> [During training together] we learned to trust each other. We came back from Strasbourg with that certain knowledge that between us we knew we would each remember something and we would be able to pull it [robot-assisted surgery] off...we seemed to develop a special bond. (Site 5, Nurse)

The underlying theory seems to be that team training works to support the integration of robot-assisted surgery into practice by establishing trust among the team. They were able to discuss the resolution of problems together, something they felt would have been impossible previously. A further benefit of training the team together was the insight it gave into the impact of the robot on other team members' roles.

## Handpicked teams

Participants perceived that team members' interest in and enthusiasm for robot-assisted surgery were enhanced when team members were handpicked to take part in whole team training. This occurred in four sites; OT personnel were handpicked by surgeons and/or nursing management to undertake robot-assisted surgery training abroad:

> It was a huge privilege to be invited...we're having this new equipment and this new concept of working and we're going to be the first people to actually really get trained properly...and then we would come back and be able to show all of the others how to do that. (Site 5, Nurse)

The underlying theory seems to be that when teams are handpicked, this creates a sense of privilege which provides staff with the motivation to overcome the challenges of robot-assisted surgery, increasing the likelihood of robot-assisted surgery becoming embedded into routine practice.

However, one participant reported that handpicking staff could have negative consequences as people resented being overlooked and consequently were not motivated to work with the robot:

> The staff that didn't go and do that training are resentful of [working with] the robot because they don't feel that they were validated enough to go and do the training abroad so why should they do the work when it's here. (Site 5, Nurse)

## Team involvement

In none of the sites had the OT team been involved in the decision to introduce robot-assisted surgery. However, in most sites, there was a positive attitude among the OT team towards robot-assisted surgery. For example, one nurse noted that, for them, there was a sense of pride as the robot added 'another string to their bow'. This view was echoed by a nurse at another site who described the robot as a 'good opportunity' in regards to their curriculum vitae and professional development.

However, attitudes at one site were notably different. While the OT team members at the other sites appeared accepting of the fact they were not involved in the decision, one nurse at Site 1 expressed disappointment about this:

> I think it's a nice piece of equipment and I would love to have been asked to be involved in making that decision, not just it being given to me, or handed to me. Because for anyone, it would be nice to have somebody to say, yes I would like to have involvement in that, it seems to be interesting to me, because that would mean they're curious and they will have that... they will be driven to learn more than if they had just been told. They can learn it more intimately than someone who has been given the job. It's something that the person made the decision to actually get involved with the robot procedures. (Site 1, Nurse)

What this quote seems to highlight is a perceived lack of control over aspects of their work; the decision they wanted to be involved in was not whether to purchase a robot, but the decision to extend use of that robot to colorectal surgery. An ODP at the same site expressed similar sentiments and felt having greater staff involvement in the decision would have positively impacted staff engagement:

> That element of communication and knowing and agreeing that this is what we're going to do from the start and this is how we're going to implement certain areas, and this is what you need, and these are the dangers and these are the benefits, and things like that. I think it's really important that the team know. And it will make them work better together, you know, you feel more comfortable if you know the bigger picture as opposed to little bits thrown in. (Site 1, ODP)

While theatre nurses and ODPs at this site expressed an appreciation of the potential benefits of robot-assisted surgery for the patient, attitudes to use of it within their hospital were generally negative. It was suggested the robot was not very popular because the team were not provided with an opportunity to learn how to use it:

We were actually kind of upset when we were told we were doing it because where was the training. We were all questioning, well I'm not trained, I wasn't particularly happy with that because I wasn't trained. I don't know I'll be safe, or my patient won't be safe when I started to do it. (Site 1, Nurse)

Thus it seems motivation to persist with robot-assisted surgery can be achieved without involvement in the initial decision to purchase a robot but training that enables team members to feel confident as they take on the new tasks is essential.

## DISCUSSION

To our knowledge, this is the first study to provide a detailed and broader-based insight into stakeholders' views of robot-assisted surgery implementation. The findings provide important information for healthcare organisations considering the introduction of robot-assisted surgery or seeking to increase use of an already purchased da Vinci robot. For such healthcare organisations, the following strategies are likely to be beneficial:

i.  Engagement of staff at different levels of the organisation: While board level support is essential for the introduction of robot-assisted surgery, it is also important to engage team leaders, as they can assist in creating conditions that accommodate the introduction of robot-assisted surgery, such as organising training and ensuring the right skill mix is available. Engagement of those surgeons who will not be using the robot is also important; if surgeons perceive the introduction of robot-assisted surgery is supported by their colleagues, they are likely to be more willing to undertake an operation with robot-assistance despite the initial longer operation duration.

ii. Handpicked dedicated robotic team: While unlikely to be feasible as a long term strategy, a handpicked dedicated team can increase the speed with which experience is built up, increasing confidence and efficiency. However, care should be taken not to alienate those who are not part of that initial team.

iii. Whole team training: Ideally the whole team should train together. This is beneficial in terms of understanding the impact of robot-assisted surgery on each other's roles, supporting teamwork.

iv. A suitably sized OT: By having a suitably sized OT, operation duration is reduced as staff are able to move quickly and the risk of de-sterilisation is reduced.

A more general issue relates to the process by which robot-assisted surgery is introduced into an organisation. The implementation of robot-assisted surgery has largely been surgeon led. This reflects a more general pattern whereby innovations are introduced into surgical practice through informal processes with an absence of quality control efforts, and some have argued this puts patients at greater risk of adverse events.[41] In none of the sites did OT team members perceive themselves to have been involved in the introduction of robot-assisted surgery. Where this is combined with a lack of training, this can create the sense that robot-assisted surgery is something thrust on the OT team, leading to feelings of resentment. While participants emphasised the importance of team leader support, it does not appear that team leaders were involved in discussions prior to the introduction of robot-assisted surgery. Our findings would suggest there is potential benefit to be gained through involving team leaders earlier in the process, so issues of training for the OT team and skill mix can be properly addressed before the robot is introduced into practice.

### Strengths and limitations

A strength of this research is that interview questions were based on analysis of existing literature, which enabled us to explore the extent to which findings from single site studies were more widely applicable. Using the theories as a starting point generated detailed accounts of contextual factors that support integration of robot-assisted surgery and how it is achieved. In addition, by interviewing the full range of OT personnel, we were able to add to and refine our literature-based theories, which came from articles predominantly authored by surgeons, to reflect the experience of a broader range of OT personnel.

A limitation of the research is that interviews were conducted with staff from only one surgical specialty, thus limiting generalisability. However, the theories explored in the interviews were derived from literature concerning a range of surgical areas. Another limitation is that while we report staff perceptions of factors that support and constrain integration of robot-assisted surgery, the resulting theories remain to be empirically tested.

### CONCLUSIONS

It is clear that the context into which robot-assisted surgery is introduced is important. Our findings suggest that, for implementation to be successful, surgeons and OT teams need to be supported at hospital and operational levels. There needs to be a culture that encourages innovation and tolerates disruption to normal practice while OT teams are learning to use the technology. A hospital which provides adequate ongoing funding, OT time and staffing may be more likely to engender and sustain enthusiasm and commitment within the team and this could lead to improved patient outcomes and safer care. Conversely, teams who feel unsupported by the hospital could become discouraged.

**Author affiliations**
[1] School of Healthcare, University of Leeds, Leeds, UK
[2] Leeds Institute of Health Sciences, University of Leeds, Leeds, UK
[3] School of Sociology and Social Policy, University of Leeds, Leeds, UK
[4] School of Management and Business, Kings College London, London, UK
[5] Centre for Decision Research, University of Leeds, Leeds, UK
[6] School of Medicine, University of Leeds, Leeds, UK
[7] School of Psychology, University of Leeds, Leeds, UK
[8] Geoffrey Giles Theatres, Leeds Teaching Hospitals NHS Trust, Leeds, UK
[9] Department of Anaesthesia, Leeds Teaching Hospitals NHS Trust, Leeds, UK

$^{10}$School of Health Sciences, University of Manchester, Manchester, UK

**Acknowledgements** The authors would like to thank the surgeons and OT personnel who generously gave up their time to be interviewed. We would like to acknowledge the support of the NIHR Clinical Research Network.

**Contributors** RR led the writing of this paper with substantial input from SH. All authors contributed to the writing of this manuscript and read and approved the final draft. RR was the overall principal investigator. RR, with JG, JH, AP, DJ, PG, AK and DD, conceived and secured funding for the study. RR, SH and NA conducted the interviews and analysed the data, with all authors contributing critically. AG contributed to the analysis of the data, providing a theatre team perspective.

**Funding** This research was funded by the National Institute for Health Research (NIHR) Health Services and Delivery Research (HS&DR) Programme (project number 12/5005/04).

**Disclaimer** The views and opinions expressed therein are those of the authors and do not necessarily reflect those of the HS&DR Programme, NIHR, NHS or the Department of Health.

**Competing interests** None declared.

**Patient consent for publication** Not required.

**Ethics approval** The University of Leeds School of Healthcare Research Ethics Committee granted ethical approval (ref: SHREC/RP/339) and participating hospitals granted research governance approval.

**Provenance and peer review** Not commissioned; externally peer reviewed.

**Data sharing statement** No additional data are available.

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
