## [Reviewer comments · BMJ Open]

ARTICLE DETAILS

TITLE (PROVISIONAL)	Factors supporting and constraining the implementation of robot-assisted surgery: a realist interview study
AUTHORS	Randell, Rebecca; Honey, Stephanie; Alvarado, Natasha; Greenhalgh, Joanne; Hindmarsh, Jon; Pearman, Alan; Jayne, David; Gardner, Peter; Gill, Arron; Kotze, Alwyn; Dowding, Dawn

VERSION 1 - REVIEW

REVIEWER	Dr Lynne Williams Bangor University Wales
REVIEW RETURNED	12-Feb-2019

GENERAL COMMENTS	Dear authors, Thank you for the opportunity to review this interesting paper. I find the paper recommendable for publication pending some minor revisions. On the whole, you manage to explain the realist approach simply and I understand why the focus in this paper is less on the methodology. However, as a reader, I'm left wondering what the tentative theories were as described (a table might be useful). I'm also not sure if "interview study" is the best way to describe this research? It might be best to affirm that this was a realist approach embedded within a process evaluation? For me, the abstract lacks a nod to the realist approach in the conclusion, and I think an opportunity is lost to highlight the approach on limiting the key words to 3. On page 7 lines 19-22 I think an opportunity is lost to explain the importance of context in the discussion about mechanisms. In 9.10 can you clarify if consent was written? In 9.10 can you elaborate to explain what is meant by cross case analysis? Was this the participant group or the hospital type? Minor points: In 8.4 -OT is not explained in full I believe. In 7.17 there is a missing word "on".
--

	In 6.10, use of the prostatectomy example was a little confusing as the study described later referred to rectal cancer. Is there an example related to rectal cancer that could be used to show the benefits of robotic surgery?
--	---

REVIEWER	Professor Michael J Solomon RPA Institute of Academic Surgery & Surgical Outcomes Research Centre University of Sydney Sydney, Australia
REVIEW RETURNED	17-Feb-2019

GENERAL COMMENTS	I must admit I am not a big fan nor an expert on "qualitative" research as it seems so random the use of "quotations" and then coming to any conclusions without any "quantitative" results to actually see how many people sampled actually agree with the statement. Usually we do the qualitative research to then design the quantitative outcome measures based on this initial work. That being said it is well done, it is certainly an established process and methodology, well written and easily understood and by submitting it to the BMJ it is locally of relevance. I think all the relevant points of starting a robotic, or any complex surgical technology for that matter, have been discussed well. I am not sure just how many of the valid points were actually discovered by the research or were a priori hypotheses though? My only criticism is the sample is very robotic proceduralists and anaesthetist "top heavy" and does not reflect the rest of the theatres who often are affected by the new technology invasion in a resource limited space? Of minor interest we have currently started and extensive robotic program solely available within a research trial framework at major teaching hospital and after 2 years of the program have currently gone straight to a quantitative survey of all theatre staff not just those involved. Look forward to contrasting these results with your study in the future and believe your qualitative study does give a baseline of attitudinal knowledge (even for a qualitative research sceptic!!). Well done.
---

VERSION 1 – AUTHOR RESPONSE

Reviewer 1

"As a reader, I'm left wondering what the tentative theories were as described (a table might be useful)."

Thank you for this suggestion. We have now added a table showing the tentative theories.

"I'm also not sure if "interview study" is the best way to describe this research? It might be best to affirm that this was a realist approach embedded within a process evaluation?"

As we describe in the methods, realist evaluation provided an overall framework for the process evaluation. The interview study was part of the first 'theory elicitation' phase of the realist evaluation, with theories then being tested through a multi-site case study in phase 2. On this basis, we consider that describing it as a realist interview study is appropriate.

"For me, the abstract lacks a nod to the realist approach in the conclusion."

We have now revised the abstract conclusion to include the following sentence: "Using a realist approach, we were also able to capture perceptions of the factors that support and constrain the integration of robot-assisted surgery into routine practice."

"I think an opportunity is lost to highlight the approach on limiting the key words to 3."

These are the keywords selected from those offered within the manuscript submission system; realist evaluation is included in our own keywords.

"On page 7 lines 19-22 I think an opportunity is lost to explain the importance of context in the discussion about mechanisms."

While we do describe the role of context ("recipients' responses to those resources... are likely to vary according to the context into which the technology is introduced"), we appreciate that the emphasis on context in realist approaches was not clear. We have now added a sentence that describes Context Mechanism Outcome configurations (in order to justify the structure of the theories in the table) which we believe makes the role of context clearer.

"In 9.10 can you clarify if consent was written?"

Interviews were undertaken by telephone so consent was verbal rather than written; we have now explained this in the manuscript.

"In 9.10 can you elaborate to explain what is meant by cross case analysis? Was this the participant group or the hospital type?"

We have revised this to indicate that by cross case we are referring to the hospital site.

"In 8.4 -OT is not explained in full I believe."

Thank you. We have corrected this.

In 7.17 there is a missing word "on".

Thank you. We have corrected this.

"In 6.10, use of the prostatectomy example was a little confusing as the study described later referred to rectal cancer. Is there an example related to rectal cancer that could be used to show the benefits of robotic surgery?"

At the time of undertaking the study, rigorous evidence of the benefits of robotic surgery in rectal cancer did not exist. The ROLARR trial, alongside which our study was undertaken, found no

advantage (but also no disadvantage) of robotic surgery over laparoscopic surgery. We have revised the sentence to make it less specific to prostatectomy, while also acknowledging that uncertainty remains regarding the benefits of robotic surgery for other types of operation.

Reviewer 2

"I am not sure just how many of the valid points were actually discovered by the research or were a priori hypotheses though?"

The inclusion of a table, showing the tentative theories that we started with from the literature, hopefully makes this clearer.

"My only criticism is the sample is very robotic proceduralists and anaesthetist "top heavy" and does not reflect the rest of the theatres who often are affected by the new technology invasion in a resource limited space?"

Forty-six per cent of our interviewees were theatre nurses and operating department practitioners, with experience of acting as both scrub practitioners and circulating practitioners.

VERSION 2 – REVIEW

REVIEWER	Dr Lynne Williams Bangor University, UK
REVIEW RETURNED	16-Apr-2019

GENERAL COMMENTS	Congratulations -this will be a very well-received paper!
---